# Corneal donation for research versus for transplantation: A-year prospective study of acceptance rates in a French University Hospital

**Thibaud Garcin**[1,2]*, **Jean Loup Pugniet**[3], **Thierry Peyragrosse**[3], **Francoise Rogues**[3], **Sophie Acquart**[4], **Fabrice Cognasse**[4], **Gilles Thuret**[1,2,5], **Philippe Gain**[1,2]

1 Corneal Graft Biology, Engineering and Imaging Laboratory, BiiGC, EA, Federative Institute of Research in Sciences and Health Engineering, Faculty of Medicine, Jean Monnet University, Saint-Etienne, France, 2 Ophthalmology Department, University Hospital, Saint-Etienne, France, 3 Hospital Coordination Team, University Hospital, Saint-Etienne, France, 4 Eye Bank, French Blood Center, Saint-Etienne, France, 5 Institut Universitaire de France, Boulevard Saint-Michel, Paris, France

* t.garcin@univ-st-etienne.fr

**Data Availability Statement:** All relevant data are within the manuscript and its Supporting Information files.

## Abstract

Fresh corneal donation is essential for basic and preclinical research, but more unknown to public and the medical teams than donation for transplantation: it may raise concerns. We prospectively compared the acceptance rates and the characteristics of targeted corneal donation for research versus donation for transplantation during one year. The Agence de la Biomédecine authorized us to procure fresh corneas targeted for research, only from the donors with medical contraindications for transplantation, in order not to increase grafts shortage. Three nurses from the hospital coordination team of Saint-Etienne University Hospital, obtained consent for research and transplantation in parallel, screening all intra-hospital deaths cases, following standard protocol to check no refusal from families, despite the French opt-out system. They contacted 127 families for research and 244 for transplantation, in 71% of cases by telephone. Consent was obtained in 62% of cases for research and 54% for transplantation (P = 0.135). The main contraindication for transplantation was the cognitive disorders (66%) followed by the blood cancers (8%). This new specific activity, providing new source of fresh corneas for research immediately usable without any eyebank storage steps, didn't reduce the number of corneas procured for transplantation versus previous years (P = 0.998). Donors in the research group were 10 years older (P<0.001) without difference regarding endothelial cell quality (P = 0.071), allowing maximal clinical relevance for protocols using these fresh human scientific corneas provided by targeted donation.

**Funding:** For development and validation of "medical device intended for long-term storage of a cornea, or for ex vivo experimentation on a human or animal cornea.": Agence Nationale de la Sécurité du Médicament (ANSM) research grant 2012 BANCO (PG); EFS 2012 research grant (GT); Jean Monnet University research grant 2014 (PG); Fondation de France Bourse Berthe Fouassier 2016 (GT); Fondation Université Jean Monnet research grant 2017 (PG); Fondation de l'Avenir Leg Deroche 2017 (GT); Agence de la Biomédecine research grant 2017 (TG). The funders had no role in study design, data collection and analysis, decision to publish, or preparation of the manuscript. P GAIN and G THURET are consultant for Thea laboratories and Quantel Medical." The funder provided support in the form of salaries for authors [PG, GT], but did not have any additional role in the study design, data collection and analysis, decision to publish, or preparation of the manuscript. The specific roles of these authors are articulated in the 'author contributions' section.

**Competing interests:** GAIN, S ACQUART and G THURET are inventors on "patent US 20160029618A1" submitted by University Jean Monnet that covers "Medical device intended for long-term storage of a cornea, or for ex vivo experimentation on a human or animal cornea." The patent does not alter our adherence to Plos One policies on sharing data and materials.

## Introduction

Organ and tissue donation is essential for fundamental and applied research. A state as close as in-vivo is equally important to ensure reliability of the results and clinical relevance. Needs are steadily increasing while availability is decreasing [1–3]. Legal and cultural differences between nations regarding donation in general and those for research complicate comparisons [4–7]. However, quantitative data are lacking as well as concrete solutions to correct imbalance.

In ophthalmology, cornea is no exception. A growing shortage of tissue for research is well documented in the USA [8, 9]. Worldwide, shortage also concerns donation for transplantation. Cornea is by far the most transplanted tissue, but in 2015 only 1/70 patients waiting worldwide could benefit from a corneal graft each year [10].

In France, corneal procurement is governed by the Bioethics Laws revised in 2018, based on the European standards [11]. Cornea is procured from a deceased donor, then necessarily stored in an authorized eyebank which carries out tissue quality and microbiological safety tests. Donation chain is an opt-out system with an online national refusal register (NRR). In absence of refusal expressed during his/her lifetime, any deceased person may legally be procured. Routinely, the hospital coordination team systematically seeks to ask relatives (without precise guidelines on degree of kinship) about the intentions of their deceased if no instruction (written or oral) is left by deceased and the relatives themselves refuse, the coordinators trust them and the procurement is never imposed.

The scientific human corneas used in France have two different origins: they come mainly from waste products of transplantation donation and a minority from body's donation to science. The first are corneas discarded by quality controls in eyebanks. They represent about 50% of 11,000 corneas procured [12], but infected corneas must be withdrawn and only those that don't have refusal for scientific use can be used. They are mostly poorer quality corneas to those required for transplantation. Besides these corneas are only available after several days of storage and are no longer fresh tissue. The second, resulting from donation in anatomy laboratories of medicine faculties, represents only a few hundred per year in France. They are the only fresh scientific corneas currently available, provided that death-to-procurement time is <24hours when body arrives in laboratory. According to legislative framework, distribution of these corneas for research are free of charge in France.

Among our research axes aims we developed an active storage machine for corneal grafts, which restores equivalent of intraocular pressure and renews storage medium. After testing it on porcine corneas [13], preclinical validation of this device was performed on a large series of human fresh corneas [14, 15]. To avoid all biases related to discarded corneas from eyebanks, we requested from the Agence de la Biomédecine (ABM, whose mission is to supervise, evaluate, promote procurement and transplantation) opportunity to procure fresh scientific human corneas, for this preclinical study and all other protocols using our device, and for protocols aiming at make evolve eyebanking, corneal imaging and bioengineering. We were concerned that donation for research could raise concerns because of its novelty and specificity. We therefore conducted a-year prospective study comparing characteristics of targeted donation for research and usual donation for transplantation.

## Material and methods

### Ethical considerations

All procedures conformed to the tenets of the Declaration of Helsinki for biomedical research involving human subjects. The ABM specifically authorized corneas procurement for research (PFS15-008 & PFS16-010) and study was approved by local Institutional Review Board "Ethics

Committee of the CHU de Saint-Etienne, Research Commission of Terre d'Ethique" (IORG0007394, N°IRBN272016/CHUSTE). The ABM asked us to select for research only persons with medical contraindication to transplantation, based on the European standards, so as not to reduce corneas number for waiting recipients (**S1 Table**). All other inclusion criteria didn't differ from those usually used for corneal transplantation donation.

## Study design

We collected prospectively during a-year, all data concerning corneal donation for research and transplantation in the Saint-Etienne university hospital. Main objective was to compare acceptance rates between both groups.

Secondary objectives were to analyze whether differences exist between both groups in donor profiles or corneas procured. We compared: 1/ consent seeking process data: obtaining consent methods (face-to-face or telephone), answer delay (immediate or delayed to think about); 2/ reasons for refusal; 3/ methods of expressing opposition to donation (NRR, instructions left to relatives); 4/ donors and corneas characteristics: endothelial cell density (ECD) measured 48hours after procurement (main quantitative quality criterion in eyebanks: done by the Saint-Etienne eyebank technicians for transplantation, and a unique skilled operator (TG) for research) and cataract operated eye (potentially with lower corneal quality); 5/ acceptance rate according to hospital coordinators experience to potentially optimize interview protocol.

Finally, to verify if this new research task impacted transplantation activity, we compared this year's study to previous 13 years' activity carried out in the same hospital environment with the same eyebank connection.

## Hospital coordination team missions

Three nurses of the hospital coordination team were involved. One was very experienced (20 years of experience and about 2000 interviews), one was experienced (15 years of experience and about 1500 interviews), the third (devoting 50% of her time to administrative-regulatory tasks) had 5 years of experience and about 500 interviews.

Regular mission consisted in obtaining consent to organ and tissue donation for transplantation. In France donation chain follows an opt-out system. More specifically for the corneas, they daily screened all intra-hospital deaths. Following French guidelines health authorities, maximum death-to-procurement time was 24 hours. Were recommended but not mandatory: death-to-body-refrigeration time <4hours, if not death-to-procurement time should be <12hours. In parallel, coordinators screened deceased persons with medical contraindication to donation for transplantation. Interviews were conducted according to standard written procedures, either face-to-face or by telephone, as we reported in 2002 [16]. For the research group, coordinators explained to the relatives of the eligible deceased that corneal donation wasn't possible for transplantation due to medical contraindication, but for local medical research: so the coordinators asked the relatives to consent to cornea donation directly for research. General purpose of the corneal grafts research axes conducted by same team of surgeons and researchers was explained, if the relatives so wished. When clarifications were required, a detailed explanatory letter from department's head (PG) could be provided to the relatives, just as follow-up support if needed. The type of the consent depended of each eligible deceased: written or oral refusal during lifetime for deceased, or oral consent or refusal (by telephone or face-to-face) from relatives contacted and informed.

## Corneas procurement

All corneas were procured by 3 trained ophthalmology residents. Fresh human corneas were procured by in situ excision with the same settings whatever the target, transferred to our laboratory within 20 minutes (for research) or to eyebank as usual (for transplantation). Costs of sterile single-use instruments and organoculture medium for the research group were covered by our research laboratory funding. The ABM authorized us to procure only fresh scientific corneas: the remaining eye tissue was left on the deceased as procurements for transplantation, with tegumentary reconstruction ad integrum.

## Statistics

Normality of continuous data distribution was analyzed with Shapiro-Wilk test with non-normality threshold of 5%. Normal distribution data were described by mean±SD, [min-max]. When the variable followed normal distribution, an unpaired Student t-test was used to compare donors' characteristics of both groups. Chi$^2$ test was used to compare percentages. Statistical significance was set at $P<0.05$, with two-tailed tests (unless specified one-tailed) and adjusted with Tukey technique when multiple tests were performed. Analyses were performed with SPSS 25.0 (IBM Corp, Armonk, NY).

# Results

## Medical wards of pre-selected deceased

In 12 months, on 1442 intra-hospital deaths, coordinators sought absence of opposition to donation for 371 deceased eligible to potential corneal donation: 127 for research, 244 (232 in circulatory arrest, 12 from multi-organ donor) for transplantation. For each group, deceased were mainly from 3 services: emergency (21%), geriatrics (20%) and intensive care (16%) for research, and intensive care (28%), emergency (12%) and pneumology (9%) for transplantation.

## Methods of interviewing the relatives and answer delay

Seventy-one percent of interviews were conducted by telephone, without difference between both groups (72% for research, 70% for transplantation P = 0.389). Answer was immediate in 69% (257/371) of cases and didn't differ between both groups (64% for research, 72% for transplantation P = 0.540). Relatives answered more often immediately during face-to-face interview (79% of cases) than by telephone (66% of cases) (P = 0.016), without difference between both groups (immediate answer was done: a/during face-to-face interview 74% for research, 81% for transplantation P = 0.393; b/ by telephone 61% for research, 68% for transplantation P = 0.218).

## Acceptance rates

Consent was obtained in 62% (79/127) of cases for research and 54% (132/244) for transplantation (P = 0.135), providing 158 fresh corneas for research and 264 for eyebank respectively. Acceptance rates weren't influenced by gender of eligible donors, with 57% (123/216) of acceptance among males and 57% (88/155) among females (P = 0.974) respectively for overall series, without difference between both groups: 62% of females gave their consent for research versus 52% for transplantation (P = 0.212); respectively 62% of males for research versus 55% for transplantation (P = 0.357). **Table 1** details acceptance rates within and between both groups, according to interview type and answer delay by relatives: no significant differences were

Table 1. The acceptance rates for the research and the transplantation groups according to interview type and answer delay.

| | | Research n = 79 | Transplantation n = 132 | P |
|---|---|---|---|---|
| Interview type | Telephone | 59% (55/93) | 50% (86/171) | 0.416 |
| | Face-to-face | 68% (24/34) | 63% (46/73) | 0.670 |
| Answer Delay | Immediate | 60% (49/82) | 51% (90/176) | 0.242 |
| | Delayed | 67% (30/45) | 62% (42/68) | 0.953 |

found. Telephone made it possible to procure 110 corneas for research and 172 for transplantation (i.e. 67% of all corneas).

Acceptance rates varied significantly with coordinator's experience: the most experienced obtained 71% and 62% of consent, respectively for research and transplantation, the one with intermediate experience 55% and 52%, the 3rd less experienced 43% and 39% (P = 0.003 for overall comparison between coordinators, both groups combined). None of them had more success for research than for transplantation (P = 0.253, P = 0.730, P = 0.824 respectively for the 3 coordinators). Coordinator's experience was directly correlated to acceptance rates whatever target: r = 0.9966 for research (P = 0.026), r = 0.9972 for transplantation (P = 0.024) (unilateral tests).

## Analysis of opposition to donation

No opposition was found on NRR. Clear (oral or written) opposition from subject prior to death, left to the relatives, was found in 19% (9/48) of cases for research versus 37.5% (42/112) for transplantation (P = 0.045). In other cases, refusal was a relatives' decision (from one person if alone, or from a collegial decision). There wasn't any influence of the gender or relationship of the person(s) providing consent (P = 0.342 and P = 0.254 respectively). Regardless target (79% for research, 80% for transplantation), the relatives mainly didn't provide any justification (P = 0.036). When refusal was mentioned, it was: "Donation isn't priority" in 10% of cases for research and 11% for transplantation (P = 0.999); "Our relative suffered too much, was too sick or too old" in 8% of cases for research and 6% for transplantation (P = 0.999); relatives didn't call coordinators back in 3% of cases for research and 3% for transplantation (P = 0.999).

## Donors characteristics and corneal cells quality

Medical contraindications that allowed selection in the research group were mainly the cognitive disorders in 66% (52/79) of cases, followed by the blood cancers (8%). The details were provided in **Fig 1**. Donors in the research group were significantly more likely to be females (P = 0.004) and older by an average of 10 years (P<0.001); their bodies were also refrigerated quicker (P<0.001). However, there wasn't significant difference in death-to-procurement time (P = 0.911), percentage of eyes with cataract surgery (P = 0.120) and ECD (P = 0.071). Twelve pairs of corneas (15%) in the research group were available for experimentation within 6 hours or less after death, and 24 (30%) within 12 hours or less, with an average death-to-experimentation in laboratory time 15h15±6hours [0h55;24h08]. Further details of these characteristics were noted in **Table 2**.

## Comparison of transplantation activity with previous years

With 244 families contacted for transplantation during this study year, activity wasn't different from that of previous 13 years, which had concerned an average of 227±24 [181 to 264]

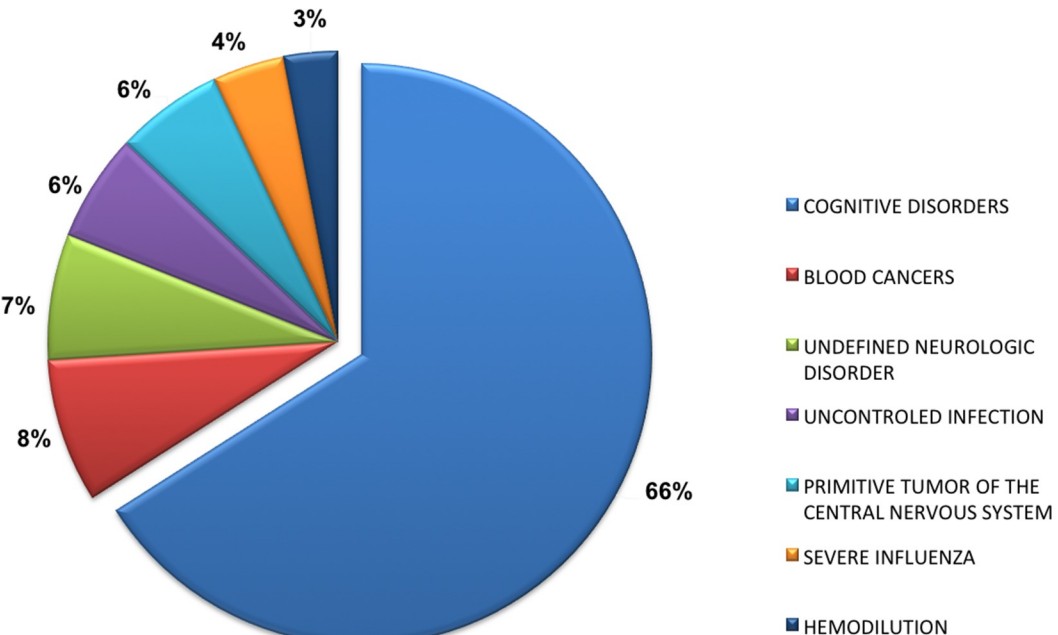

**Fig 1. The medical contraindications that allowed procurement in the research group.** Note that cognitive disorders, first source of targeted corneas for research, include any neurodegenerative disease (such as Alzheimer, Parkinson disease, Multiple Sclerosis, Amyotrophic lateral sclerosis...). These diseases represent contraindications for corneal procurement and transplantation, to ensure safety of the graft: the principle is to rule out any degenerative disease presumed to be transmissible via corneal transplantation. These contraindications follow a precautionary principle.

families (P = 0.994). Similarly, number of corneas procured for transplantation (264) didn't differ from the average of previous 13 years (284±24 [from 223 to 330] (P = 0.998)).

## Discussion

Corneal donation for research is still too little-known to public and must be encouraged as much as that for transplantation, both of which are essential to fight against blindness. We showed that it is possible to specifically target deceased to procure fresh corneas for research, in parallel with transplantation activity, without putting them in competition.

Acceptance rate didn't differ between the research and the transplantation groups, whereas we feared that donation for research might have raised more reluctance [17, 18]. Several mutual and non-exclusive explanations can be formulated: 1/ people who accept idea of corneal donation don't oppose research and transplantation; 2/ some may think that they help even more people via research [19] than transplantation, which treats traditionally only two

**Table 2. The characteristics of donors and corneas procured in the research and the transplantation groups.** Continuous data are expressed in mean±SD [min-max]; ECD = endothelial cell density.

|  | Research n = 79 | Transplantation n = 132 | P |
|---|---|---|---|
| Sex ratio Male/Female | 0.84 (36 M/43 F) | 1.93 (87 M/45 F) | 0.004 |
| Donor age (years) | 81±12 [49;100] | 71±13 [25;101] | <0.001 |
| Death-to-refrigeration time (hour) | 3h36±2 [1h30;11h20] | 7h32±1 [1h00;20h39] | <0.001 |
| Death-to-procurement time (hour) | 15h05±6 [0h45;23h58] | 15h01±6 [3h00;23h50] | 0.911 |
| % of eyes with cataract surgery | 30% (48/158) | 22% (58/264) | 0.120 |
| Average ECD (cells/mm$^2$) | 2540±479 | 2641±596 | 0.071 |

patients; 3/ older age (10 years) in the research group can make higher consent rate [20]; 4/ great experience of the coordinators in presenting legislation and research issues with empathy plays a key role [3, 21, 22]; 5/ longstanding [16] involvement of the coordinators in research projects, directly related here to improve corneal grafts storage keeps motivation over time; 6/ general public's knowledge of our health region and team's research works (dedicated mainly but not only to corneal eyebanking and transplantation) through regular follow-up support for the relatives may help to make meaning of donation. But in practice relatives didn't ask the coordinators for so much details about research aims.

Our high acceptance rate for transplantation was significantly higher than those reported for transplantation in literature [18, 21, 23] where rates exceeding two-thirds acceptance are exceptional [16, 24, 25]. Comparison between the teams is highly complex because it questions donation regulation, overall organization of the teams, donors screening and experience of the coordinators evolving over time. Our findings result certainly from combination of an opt-out system and a small well-structure, highly motivated experienced team. Key role of "human factor" in success of this deeply personal approach was reported before [25–27].

Our study further confirms crucial role of telephone interview, reported regularly [20, 28, 29]. In 2002 [16], we showed that for transplantation, telephone accounted for 58% of interviews and 48% of the corneas procured. Here for transplantation, its role is even more important with 71% of interviews and 67% of corneas procured. While in 2002 telephone was less efficient than face-to-face interviews, this difference wasn't significant in 2017: communication resources evolution with the smartphones pervasive in daily life, is probably involved.

Refusals analysis shows that NRR is probably widely underused. The donation opponents tell probably rarely their relatives why, with a potential memory bias, as the justifications collected by our coordinators seem to indicate. Whatever target, donation legislative framework like presence of NRR aren't well known to general public. Generally speaking, question of donation isn't still sufficiently addressed in families, even for favorable people with only 50% of them having informed their relatives [22, 30]. At time of death, when relatives have to decide in place of their deceased, fear of doing wrong or assuming responsibility can lead to refusal.

The medical contraindications to corneal donation for transplantation are numerous, based mainly on precautionary principle solely and deprive recipients of potentially great intrinsic quality corneas: only rabies (constantly fatal), herpes simplex virus, bacteria, fungi and retinoblastoma have been formally demonstrated to be transmissible via the cornea. So, targeting these contraindications allows researchers to benefit from them. Some sporadic cases of prion disease were reported in recipients, and circumstantial evidence has implicated corneal transplantation as a mechanism of transmission of iatrogenic prion disease: transplantation is presumed (but not formal) to be the source of prion disease in recipients [31–34]. Although we could be concern about disease transmission or research bias induced by these contraindications, it clearly depends on the research aim(s). In this study, every cornea was procured by in situ excision with single-use sterile instrument, without any contact with retina or optic nerve, which are be considered as specific risk factors to possible iatrogenic spread of sporadic and variant prion disease [35, 36]. We did risk analysis before the use of those corneas to develop our active storage machine to make evolve eyebanking, with assessment of their baseline intrinsic quality (endothelial cell density [37–40], transparency [41], presence of scar or not, presence of previous refractive surgery or not) and safety (no infection [42–44]), following tests used in daily routine in eyebanks. Furthermore, with these common medical contraindications people may wish to give but feel excluded from donation: donation for research can be a way to give them back this opportunity. However many people are interested in

donating their corneas for research but aren't aware that it was possible to do so [45]. Thus, they leave more often no instruction to their relatives concerning procurement for research.

Targeted donation for research has another advantage: it allows the researchers to dispose the fresh tissues immediately after procurement, without passing through the eyebanks. Several positive consequences can be stated: it provides short circuit, without extra-work for the eyebanks and less charges for the researchers at the end of the chain (variations occur in different countries, such as the USA where procurements are done by paid technicians); it removes the damaging processes on cornea quality induced by storage itself.

Our specific approach imposed by the ABM made it possible to procure a large quantity of fresh scientific corneas over a short period of time, without difference quality from those grafted, ensuring optimal clinical relevance for the research protocols in eyebanking. Through people giving their bodies to science, we found that we could just procure 41±17 fresh scientific corneas per year over last 3 years in our faculty of medicine: it would have taken almost 4 times longer to obtain same number of fresh corneas versus our study with 158 corneas a-year. Without these fresh tissues, preclinical validation of our active storage machine [14] couldn't have been performed under same conditions as its future use, like the other following protocols to improve corneal storage using our device with the extensions of authorizations by the ABM. Death-to-refrigeration time difference between groups was explained by difference in the services from which deceased came, not involving same post-death logistics. Donors age was logically higher, with more females in the research group, since we were asked to select potential donors for research on the diseases that were more frequent with aging and females live classically longer. It didn't impact research protocols, since donor age isn't a significant factor in survival of corneal grafts [46, 47].

Concerning coordinators, despite their additional workload to obtain targeted donation for research, there wasn't negative impact on already important transplantation activity, neither on number of the families interviewed nor on number of corneas procured. That underlines possibility to have clinical and research processes coordinated without penalizing each other. The coordinators increased their efficiency without additional working time, didn't receive bonus, made and have pursued it with belief: whatever research and clinical endeavors, they did both tasks without any conflict. Donation chain may vary in different countries. Regarding cost effectiveness of targeted corneal donation, we believe it may be applied whatever medico-economic system. In France, our laboratory had to pay only materials. But if needed, extra-cost from people in charge of obtaining consent or/and procurements, must be implemented at chain's end for researchers. Anyway, without passing through the eyebanks, researchers will have less fees and better-quality fresh corneas, immediately available for experimentations, with this targeted corneal donation.

Despite being prospective and designed to minimize potential bias, our study presents some limitations: 1/ center-effect with highly motivated coordinators that have strong and long-lasting links with medical-research team for two decades. However, we believe that extrapolation to the other research teams motivated to obtain exceptional quality tissues, could have similar results. 2/ net inter-coordinator effect. Despite using standard protocol, some variations were intrinsically linked to age-related experience: more experienced coordinator probably adapts better to each family profile [26, 28] whatever target donation.

Several proposals have recently been made to improve acceptance of corneal donation for research. In the USA, Williams *et al.* advocates for scientific donation [9], by proposing creation of an eye donation registry for research [48]; collaborations between the eyebanks and the research institutes to recover corneas unsuitable for transplantation [49], as we have implemented since almost 20 years to procure discarded grafts from our eyebank. Besides, [50] proposed the creation of an online portal, under the aegis of the Association for Research in

Vision and Ophthalmology, to specifically link the eyebanks to researchers in need of eye tissue.

To make this work useful to other teams, we may suggest some ways to increase number of donation for research: 1/ to develop privileged relationships between the research laboratories and the team in charge of donation in local hospital. Exposing her regularly aims, advances and results of researches, is probably crucial to induce lasting links; 2/ to encourage companionship between the novice and the experienced coordinators; 3/ to integrate in donation promotion campaigns, awareness raising to donation legislative framework and possibility to donate for research; 4/ to increase the resources allocated to the coordinators and ensure that all eligible relatives of deceased can be contacted in a timely manner: pressure comes from legislative framework and bodies transportation from hospital mortuary to private funeral homes, explaining mainly gap between number of intrahospital deaths and deceased eligible contacted in this study. Respectively 413 families were not contacted due to legislative framework with death-to-procurement time >24h (which discards deceased from eligibility to for corneal donation for research or transplantation); 411 families were not contacted due to fast transport of the deceased body from hospital mortuary to private funeral homes or conservative care (which prevent from interviewing families for research or transplantation). The third cause is that sometimes the coordinators didn't have enough time to treat every dossier of eligible deceased (n = 243 families) for research or for transplantation in their daily multi-tasks: they share time between corneas, other tissues (vessels, bone), organs (mainly kidney).

In summary, in parallel with transplantation activity, by targeting donors with medical contraindication to corneal donation for transplantation, we can obtain many fresh scientific corneas of similar quality to those grafted, immediately available for the researchers, without increasing shortage for waiting recipients or incurring extra-fees by passing through the eyebanks. This targeted corneal donation could be a potential solution to make research advance better and faster.

## Supporting information

**S1 Table. List of the medical contraindications to corneal donation for transplantation in effect at the time of the study (2017).**
(DOCX)

## Acknowledgments

We are grateful to those who donated their corneas to science, and to their families. We also thank the Agence de la Biomédecine for its institutional support and the authorizations.

## Author Contributions

**Conceptualization:** Thibaud Garcin, Jean Loup Pugniet, Gilles Thuret, Philippe Gain.

**Data curation:** Thibaud Garcin, Jean Loup Pugniet, Thierry Peyragrosse, Francoise Rogues.

**Formal analysis:** Thibaud Garcin, Thierry Peyragrosse, Francoise Rogues, Sophie Acquart, Gilles Thuret.

**Funding acquisition:** Gilles Thuret, Philippe Gain.

**Investigation:** Thibaud Garcin, Thierry Peyragrosse, Philippe Gain.

**Methodology:** Thibaud Garcin, Jean Loup Pugniet, Francoise Rogues, Sophie Acquart, Fabrice Cognasse, Gilles Thuret.

**Resources:** Thibaud Garcin, Jean Loup Pugniet, Thierry Peyragrosse, Francoise Rogues, Fabrice Cognasse, Gilles Thuret.

**Software:** Thibaud Garcin, Thierry Peyragrosse, Gilles Thuret.

**Supervision:** Thibaud Garcin, Jean Loup Pugniet, Gilles Thuret, Philippe Gain.

**Validation:** Thibaud Garcin, Thierry Peyragrosse, Francoise Rogues, Sophie Acquart, Fabrice Cognasse, Gilles Thuret, Philippe Gain.

**Visualization:** Thibaud Garcin, Gilles Thuret, Philippe Gain.

**Writing – original draft:** Thibaud Garcin.

**Writing – review & editing:** Thibaud Garcin, Gilles Thuret, Philippe Gain.

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
