## [Decision Letter · Decision Letter 0]

13 Feb 2020

PONE-D-19-35417

Corneal donation for research versus for transplantation: a-year prospective study of acceptance rates in a French University Hospital

PLOS ONE

Dear Dr. GARCIN,

Thank you for submitting your manuscript to PLOS ONE. After careful consideration, we feel that it has merit but does not fully meet PLOS ONE’s publication criteria as it currently stands. Therefore, we invite you to submit a revised version of the manuscript that addresses the points raised during the review process.

We would appreciate receiving your revised manuscript by Mar 29 2020 11:59PM. To enhance the reproducibility of your results, we recommend that if applicable you deposit your laboratory protocols in protocols.io, where a protocol can be assigned its own identifier (DOI) such that it can be cited independently in the future. For instructions see: http://journals.plos.org/plosone/s/submission-guidelines#loc-laboratory-protocols

We look forward to receiving your revised manuscript.

Kind regards,

Yu-Chi Liu, M.D

Academic Editor

PLOS ONE

Journal Requirements:

2. Please provide additional details regarding participant consent. In the Methods section, please ensure that you have specified (1) whether consent was informed and (2) what type you obtained (for instance, written or verbal). If your study included minors, state whether you obtained consent from parents or guardians. If the need for consent was waived by the ethics committee, please include this information.

3. Thank you for including your ethics statement: The Agence de la Biomedecine (ABM) specifically authorized corneas procurement for research (PFS15-008 & PFS16-010) and study was approved by local Institutional Review Board (IORG0007394, N°IRBN272016/CHUSTE).

4. We note that you have a patent relating to material pertinent to this article. Please provide an amended statement of Competing Interests to declare this patent (with details including name and number), along with any other relevant declarations relating to employment, consultancy, patents, products in development or modified products etc. Please confirm that this does not alter your adherence to all PLOS ONE policies on sharing data and materials, as detailed online in our guide for authors http://journals.plos.org/plosone/s/competing-interests by including the following statement: "This does not alter our adherence to  PLOS ONE policies on sharing data and materials.” If there are restrictions on sharing of data and/or materials, please state these. Please note that we cannot proceed with consideration of your article until this information has been declared.

"P GAIN, S ACQUART and G THURET are inventors on “patent US 20160029618A1” submitted by University Jean Monnet that covers “Medical device intended for long-term storage of a cornea, or for ex vivo experimentation on a human or animal cornea”.

P GAIN and G THURET are consultant for Thea laboratories and Quantel Medical."

We note that one or more of the authors are employed by a commercial company: Thea laboratories and Quantel Medical.

Reviewers' comments:

Reviewer's Responses to Questions

**Comments to the Author**

1. Is the manuscript technically sound, and do the data support the conclusions?

Reviewer #1: Yes

Reviewer #2: Yes

2. Has the statistical analysis been performed appropriately and rigorously? 

Reviewer #1: Yes

Reviewer #2: Yes

3. Have the authors made all data underlying the findings in their manuscript fully available?

Reviewer #1: Yes

Reviewer #2: Yes

4. Is the manuscript presented in an intelligible fashion and written in standard English?

Reviewer #1: Yes

Reviewer #2: Yes

5. Review Comments to the Author

Reviewer #1: The study is technically sound with a valid scientific question. The results were interesting and surprising. We note that the paper did not mention whether the research cornea donors were tested for relevant communicable diseases and what the disposition of the corneas would be should the results come back positive. Was there any concern that some of the cognitive disorders were due to prion disease?

Our other query would be the consent taking process for biomedical research, perhaps this can be mentioned in the paper? Will this be different from the consent for clinical transplant purposes? Or can we assume that ABM has already provided the necessary authorization and therefore no further consent is required?

The manuscript is easily understood but there is room for improvement with regard to overall sentence construction, grammatical flow and word choices.

Agree with the author’s view that the high consent rates may be a single centre effect and may not represent the situation in most other institutions around the world. However, it echoes the need for a consistent and sustained messaging to the general public about cornea donation, whether it be for therapy, research or education.

Reviewer #2: In this prospective study, the authors compared the donation acceptance rate for cornea transplantation (deceased patients with no medical contraindication for corneal transplantation) and for research (deceased patients with contraindication for corneal transplantation) and evaluated if the acceptance rate for transplantation donation remained stable over a year.

General comments

Obtaining fresh corneas for research is highly challenging and a main concern for worldwide researchers. While it is highly interesting to see that acceptance rates for research and for transplant were comparable, this paper opens several questions regarding research regulations of corneas. Indeed, in France, historically, ineligible corneas for transplant were being used for research, in consequence, research corneas came from patients without medical contraindication for transplant.

In the present study, corneas from deceased patients with medical contraindication for transplant (including cognitive disorders, blood cancers, neurologic disorder, uncontrolled infection, tumor of CNS, severe influenza and hemodilution) were used for research to increase the number of fresh corneas needed for research purpose.

The main question here might be “should we use tissues from cognitive disorders, blood cancers, neurologic disorder, tumor of CNS, uncontrolled infections… for our research?” Indeed, the authors explained that one of the main focus of their lab was to develop an active storage machine for corneal graft. We could be concern about both disease transmission and induced research bias which is not at all discussed in the paper.

Thus, the discussion should address those points which can’t be only resumed in one sentence (Line 279-281) “The medical contraindications to corneal donation for transplantation are numerous, based mainly on precautionary principle solely and deprive recipients of potentially great intrinsic quality corneas”.

Was any risk analysis performed before the use of those corneas?

Point by point comments

Some additional points need to be clarified and are listed below.

Results

Line 153 the authors reported 1442 intra-hospital deaths but only 371 deceased eligible. Can the authors explain the reasons of the gap since cornea for research are based on medical contraindications meaning that families of all deceased patients should be interviewed. What were the limitations?

Table 1 is nonreadable, please correct. In addition, in the table, the total of research interviews is 126 but number written in results line 154 is 127. Please review and correct.

Line 212 “medical contraindications that allowed selection in the research group” can you please clarify the French rules about research on tissues with medical contraindications for transplant?

End of comments

6. PLOS authors have the option to publish the peer review history of their article (what does this mean?). If published, this will include your full peer review and any attached files.

Reviewer #1: Yes: Howard Cajucom -Uy

Reviewer #2: No

---

## [Author Response · Author response to Decision Letter 0]

9 Mar 2020

PONE-D-19-35417

Corneal donation for research versus for transplantation: a-year prospective study of acceptance rates in a French University Hospital

• A rebuttal letter that responds to each point raised by the academic editor and reviewer(s). This letter should be uploaded as separate file and labeled 'Response to Reviewers'.

• A marked-up copy of your manuscript that highlights changes made to the original version. This file should be uploaded as separate file and labeled 'Revised Manuscript with Track Changes'.

• An unmarked version of your revised paper without tracked changes. This file should be uploaded as separate file and labeled 'Manuscript'.

First, we would like to thank the Academic Editor, reviewer #1 and #2 for her/his reviewing and advises. We have chosen to respond to the comments and to resubmit this manuscript. We deeply apologize for grammar or language mistakes.

Journal Requirements:

 No problem, we followed guidelines

2. Please provide additional details regarding participant consent. In the Methods section, please ensure that you have specified (1) whether consent was informed and (2) what type you obtained (for instance, written or verbal). If your study included minors, state whether you obtained consent from parents or guardians. If the need for consent was waived by the ethics committee, please include this information.

Consent was informed. In France we have opt out system. The type of the consent depended of each eligible deceased : written or oral refusal during lifetime for deceased, or oral consent or refusal (by phone or face to face) from relatives contacted.

We precised it in Methods section.

3. Thank you for including your ethics statement: The Agence de la Biomedecine (ABM) specifically authorized corneas procurement for research (PFS15-008 & PFS16-010) and study was approved by local Institutional Review Board (IORG0007394, N°IRBN272016/CHUSTE).

We have amended the full name of our institutional review board that approved our specific study Ethics Committee of the CHU de Saint-Etienne, Research Commission of Terre d’Ethique

We precised it in the Methods Section and in the “Ethics statement” field of the submission form.

4. We note that you have a patent relating to material pertinent to this article. Please provide an amended statement of Competing Interests to declare this patent (with details including name and number), along with any other relevant declarations relating to employment, consultancy, patents, products in development or modified products etc. Please confirm that this does not alter your adherence to all PLOS ONE policies on sharing data and materials, as detailed online in our guide for authors http://journals.plos.org/plosone/s/competing-interests by including the following statement: "This does not alter our adherence to PLOS ONE policies on sharing data and materials.” If there are restrictions on sharing of data and/or materials, please state these. Please note that we cannot proceed with consideration of your article until this information has been declared.

 The patent does not alter our adherence to PLOS ONE policies on sharing data and materials.

This information should be included in your cover letter; we will change the online submission form on your behalf. OK no problem.

"P GAIN, S ACQUART and G THURET are inventors on “patent US 20160029618A1” submitted by University Jean Monnet that covers “Medical device intended for long-term storage of a cornea, or for ex vivo experimentation on a human or animal cornea”.

P GAIN and G THURET are consultant for Thea laboratories and Quantel Medical."

We note that one or more of the authors are employed by a commercial company: Thea laboratories and Quantel Medical.

Please also include the following statement within your amended Funding Statement. OK, no problem.

“The funder provided support in the form of salaries for authors [PG, GT], but did not have any additional role in the study design, data collection and analysis, decision to publish, or preparation of the manuscript. The specific roles of these authors are articulated in the ‘author contributions’ section.”

 No commercial affiliation played role in our study.

Commercial affiliations do not alter our adherence to PLOS ONE policies on sharing data and materials.

We apologize for that, we precised this point in reviewed manuscript.

Reviewers' comments:

Reviewer's Responses to Questions

Comments to the Author

1. Is the manuscript technically sound, and do the data support the conclusions?

Reviewer #1: Yes

Reviewer #2: Yes 

2. Has the statistical analysis been performed appropriately and rigorously? 

Reviewer #1: Yes

Reviewer #2: Yes 

3. Have the authors made all data underlying the findings in their manuscript fully available?

Reviewer #1: Yes

Reviewer #2: Yes 

4. Is the manuscript presented in an intelligible fashion and written in standard English?

Reviewer #1: Yes

Reviewer #2: Yes 

5. Review Comments to the Author

Reviewer #1: The study is technically sound with a valid scientific question. The results were interesting and surprising. We note that the paper did not mention whether the research cornea donors were tested for relevant communicable diseases and what the disposition of the corneas would be should the results come back positive. Was there any concern that some of the cognitive disorders were due to prion disease? 

Thank you for your comment, we have added the following sentences in the discussion section (lines 291 to 307)

“only rabies (constantly fatal), herpes simplex virus, bacteria, fungi and retinoblastoma have been formally demonstrated to be transmissible via the cornea. So, targeting these contraindications allows researchers to benefit from them. Some sporadic cases of prion disease were reported in recipients, and circumstantial evidence has implicated corneal transplantation as a mechanism of transmission of iatrogenic prion disease : transplantation is presumed (but not formal) to be the source of prion disease in recipients [31-34]. Although we could be concern about disease transmission or research bias induced by these contraindications, it clearly depends on the research aim(s). In this study, every cornea was procured by in situ excision with single-use sterile instrument, without any contact with retina or optic nerve, which are be considered as specific risk factors to possible iatrogenic spread of sporadic and variant prion disease [35, 36].”

Our other query would be the consent taking process for biomedical research, perhaps this can be mentioned in the paper? Will this be different from the consent for clinical transplant purposes? Or can we assume that ABM has already provided the necessary authorization and therefore no further consent is required? 

Thank you for your comment. We clarified this point lines 122 to 139.

The manuscript is easily understood but there is room for improvement with regard to overall sentence construction, grammatical flow and word choices. We deeply apologize for grammar or language mistakes. We updated this point.

Agree with the author’s view that the high consent rates may be a single centre effect and may not represent the situation in most other institutions around the world. However, it echoes the need for a consistent and sustained messaging to the general public about cornea donation, whether it be for therapy, research or education.

Reviewer #2: In this prospective study, the authors compared the donation acceptance rate for cornea transplantation (deceased patients with no medical contraindication for corneal transplantation) and for research (deceased patients with contraindication for corneal transplantation) and evaluated if the acceptance rate for transplantation donation remained stable over a year.

General comments

Obtaining fresh corneas for research is highly challenging and a main concern for worldwide researchers. While it is highly interesting to see that acceptance rates for research and for transplant were comparable, this paper opens several questions regarding research regulations of corneas. Indeed, in France, historically, ineligible corneas for transplant were being used for research, in consequence, research corneas came from patients without medical contraindication for transplant.

In the present study, corneas from deceased patients with medical contraindication for transplant (including cognitive disorders, blood cancers, neurologic disorder, uncontrolled infection, tumor of CNS, severe influenza and hemodilution) were used for research to increase the number of fresh corneas needed for research purpose.

The main question here might be “should we use tissues from cognitive disorders, blood cancers, neurologic disorder, tumor of CNS, uncontrolled infections… for our research?” Indeed, the authors explained that one of the main focus of their lab was to develop an active storage machine for corneal graft. We could be concern about both disease transmission and induced research bias which is not at all discussed in the paper. Thanks for this comment. We added new comment concerning this point. (cf. response to comment of reviewer #1 about prion disease => line 291 to 307)

Thus, the discussion should address those points which can’t be only resumed in one sentence (Line 279-281) “The medical contraindications to corneal donation for transplantation are numerous, based mainly on precautionary principle solely and deprive recipients of potentially great intrinsic quality corneas”. We added new comment concerning this point (line 291 to 307). 

Was any risk analysis performed before the use of those corneas? Yes, We added new comment concerning this point (line 303 to 307).

“We did risk analysis before the use of those corneas to develop our active storage machine to make evolve eyebanking, with assessment of their baseline intrinsic quality (endothelial cell density, transparency, presence of scar or not, presence of previous refractive surgery or not) and safety (no infection), following tests used in daily routine in eyebanks.”

Point by point comments

Some additional points need to be clarified and are listed below.

Results

Line 153 the authors reported 1442 intra-hospital deaths but only 371 deceased eligible. Can the authors explain the reasons of the gap since cornea for research are based on medical contraindications meaning that families of all deceased patients should be interviewed. What were the limitations? , We added new comment concerning this point (line 371 to 380).

Gap can be explained by several causes : 

1/ legislative framework with death-to-procurement time >24h, which discards deceased from eligibility to for corneal donation for research or transplantation (n= 413) 

2/ fast transport of the deceased body from hospital mortuary to private funeral homes or conservative care, which prevent from interviewing families for research or transplantation (n=411)

3/ Only 3 nurses of the coordination team are present to do all the work about corneas but also for other tissues (vessels, bone) and organs (mainly kidney). So, sometimes they do not have enough time to treat every dossier of eligible deceased for research or for transplantation in their daily tasks (between corneas, other tissues, organs) (n=243) ; so increase the resources allocated to the coordinators is crucial.

4/ medico-legal impediment with prosecutor objection (n=4).

Table 1 is nonreadable, please correct. In addition, in the table, the total of research interviews is 126 but number written in results line 154 is 127. Please review and correct. We apologize for that ; we reviewed and corrected on revised manuscript. 

Line 212 “medical contraindications that allowed selection in the research group” can you please clarify the French rules about research on tissues with medical contraindications for transplant? 

The Rules are edited and updated regularly by the Health Authorities, and this is The Agence de la biomédecine provides the rules : for each tissue medical contraindications exist for transplantation. Ineligibility for transplantation makes tissue discarded and so destructed. But some of this discarded tissue which does not represent infectious risks, may be used by specific, certified and authorized laboratories. For ocular tissue, distribution of tissue are free of charge in France.

End of comments

6. PLOS authors have the option to publish the peer review history of their article (what does this mean?). If published, this will include your full peer review and any attached files.

Do you want your identity to be public for this peer review? For information about this choice, including consent withdrawal, please see our Privacy Policy.

Reviewer #1: Yes: Howard Cajucom -Uy

Reviewer #2: No

---

## [Decision Letter · Decision Letter 1]

30 Apr 2020

PONE-D-19-35417R1

Corneal donation for research versus for transplantation: a-year prospective study of acceptance rates in a French University Hospital

PLOS ONE

Dear Dr. GARCIN,

Thank you for submitting your manuscript to PLOS ONE. After careful consideration, we feel that the manuscript has significantly improved but the reviewers still have some minor concerns. Therefore, we invite you to submit a revised version of the manuscript that addresses the points raised during the review process.

We would appreciate receiving your revised manuscript by Jun 14 2020 11:59PM. To enhance the reproducibility of your results, we recommend that if applicable you deposit your laboratory protocols in protocols.io, where a protocol can be assigned its own identifier (DOI) such that it can be cited independently in the future. For instructions see: http://journals.plos.org/plosone/s/submission-guidelines#loc-laboratory-protocols

We look forward to receiving your revised manuscript.

Kind regards,

Yu-Chi Liu, M.D

Academic Editor

PLOS ONE

Reviewers' comments:

Reviewer's Responses to Questions

**Comments to the Author**

1. If the authors have adequately addressed your comments raised in a previous round of review and you feel that this manuscript is now acceptable for publication, you may indicate that here to bypass the “Comments to the Author” section, enter your conflict of interest statement in the “Confidential to Editor” section, and submit your "Accept" recommendation.

Reviewer #1: All comments have been addressed

Reviewer #2: (No Response)

2. Is the manuscript technically sound, and do the data support the conclusions?

Reviewer #1: Yes

Reviewer #2: Partly

3. Has the statistical analysis been performed appropriately and rigorously? 

Reviewer #1: Yes

Reviewer #2: Yes

4. Have the authors made all data underlying the findings in their manuscript fully available?

Reviewer #1: Yes

Reviewer #2: Yes

5. Is the manuscript presented in an intelligible fashion and written in standard English?

Reviewer #1: Yes

Reviewer #2: Yes

6. Review Comments to the Author

Reviewer #1: (No Response)

Reviewer #2: Discussion

Line 268 misspelling of “iin” please replace by “in”

The authors may want to add some additional information regarding the CoVid current situation which will probably change their future practice. In their study, all deceased patients were eligible for research donation, will they now consider testing all deceased patients before harvesting corneas? Are French rules currently changing regarding cornea donation for research?

7. PLOS authors have the option to publish the peer review history of their article (what does this mean?). If published, this will include your full peer review and any attached files.

Reviewer #1: Yes: Howard Cajucom-Uy

Reviewer #2: No

---

## [Author Response · Author response to Decision Letter 1]

1 May 2020

PONE-D-19-35417R1

Corneal donation for research versus for transplantation: a-year prospective study of acceptance rates in a French University Hospital

PLOS ONE

Saint-Etienne, May 1st 2020

Dear Professor Yu-Chi Liu,

Please find attached a revised version of our article titled “Corneal donation for research versus for transplantation: a-year prospective study of acceptance rates in a French University Hospital”.

We have carefully considered and responded to all the points addressed by the reviewers.

Per your instructions, all substantive amendments in the revised version are stated in our point-by-point response, and are marked in red in the article. 

We greatly hope that this new version will meet the reviewers' expectations and comply with your editorial policy. 

Yours sincerely,

Dr. Thibaud GARCIN, M.D., Ph.D., FEBO

Saint-Etienne University Hospital

"Corneal Graft Biology, Engineering, and Imaging" Laboratory EA 2521

Faculty of Medicine

Saint-Etienne

France

Dear Dr. GARCIN,

Thank you for submitting your manuscript to PLOS ONE. After careful consideration, we feel that the manuscript has significantly improved but the reviewers still have some minor concerns. Therefore, we invite you to submit a revised version of the manuscript that addresses the points raised during the review process.

We would appreciate receiving your revised manuscript by Jun 14 2020 11:59PM. To enhance the reproducibility of your results, we recommend that if applicable you deposit your laboratory protocols in protocols.io, where a protocol can be assigned its own identifier (DOI) such that it can be cited independently in the future. For instructions see: http://journals.plos.org/plosone/s/submission-guidelines#loc-laboratory-protocols

• A rebuttal letter that responds to each point raised by the academic editor and reviewer(s). This letter should be uploaded as separate file and labeled 'Response to Reviewers'.

• A marked-up copy of your manuscript that highlights changes made to the original version. This file should be uploaded as separate file and labeled 'Revised Manuscript with Track Changes'.

• An unmarked version of your revised paper without tracked changes. This file should be uploaded as separate file and labeled 'Manuscript'.

We look forward to receiving your revised manuscript.

Kind regards,

Yu-Chi Liu, M.D

Academic Editor

PLOS ONE

Reviewers' comments:

Reviewer's Responses to Questions

Comments to the Author

1. If the authors have adequately addressed your comments raised in a previous round of review and you feel that this manuscript is now acceptable for publication, you may indicate that here to bypass the “Comments to the Author” section, enter your conflict of interest statement in the “Confidential to Editor” section, and submit your "Accept" recommendation.

Reviewer #1: All comments have been addressed

Reviewer #2: (No Response)

2. Is the manuscript technically sound, and do the data support the conclusions?

Reviewer #1: Yes

Reviewer #2: Partly 

3. Has the statistical analysis been performed appropriately and rigorously? 

Reviewer #1: Yes

Reviewer #2: Yes

4. Have the authors made all data underlying the findings in their manuscript fully available?

Reviewer #1: Yes

Reviewer #2: Yes

5. Is the manuscript presented in an intelligible fashion and written in standard English?

Reviewer #1: Yes

Reviewer #2: Yes

6. Review Comments to the Author

Reviewer #1: (No Response)

Reviewer #2: Discussion

Line 268 misspelling of “iin” please replace by “in”

This has been corrected

The authors may want to add some additional information regarding the CoVid current situation which will probably change their future practice. In their study, all deceased patients were eligible for research donation, will they now consider testing all deceased patients before harvesting corneas? Are French rules currently changing regarding cornea donation for research?

Thank you for the suggestion. The continuation of corneal procurement as early as possible is indeed crucial both for patients waiting for transplants and for laboratories like ours that work on these irreplaceable human tissues. The SARS-Cov-2 epidemic has effectively stopped all but multi-organ donation. It appears that we may soon be able to resume therapeutic procurement from symptomatic, non-at-risk individuals, but there are many unanswered questions. 

In order to be able to answer the question of the risk of donor-recipient transmission, we have already started a new research work and obtained the authorization from our health authority (Biomedicine Agency, PFS-2020-011) to resume retrieval for scientific purposes for the next 150 donors. Thus, we will systematically collect and test all donors (nasopharynx, conjunctiva and cornea) and carry out serologies. This will provide us with robust data to make good decisions about what new tests to perform or not to perform on donors, for transplantation and for research. We hope that the new recommendations will be based on evidence and not on an unfounded precautionary principle.

Of course, during this new study we are recording the acceptance rate for corneal donation among all potential donors, and possible reasons for refusal. We will know whether the current epidemic is changing behavior and we will submit a letter to report the results, if you wish.

For this article, as suggested, we would like to add the following paragraph as the end of discussion:

“By the time this article is accepted, the global SARS-Cov-2 epidemic has stopped corneal procurement for therapeutic and scientific purposes altogether. Scientific knowledge is sorely lacking to establish the risk of transmission of this virus via ocular tissues and therefore to make new recommendations on which virological tests should be added or not to ensure total safety. Our French health authority has authorized us in emergency (PFS-2020-011) to perform a large series of scientific procurements from any potential donor (COVID+ or -) in order to objectively analyze the risks. During this particular study we will also analyze whether the epidemic has modified the acceptance rate and the reasons for a possible refusal.”

Respectfully

Dr T Garcin, MD, PhD, FEBO

7. PLOS authors have the option to publish the peer review history of their article (what does this mean?). If published, this will include your full peer review and any attached files.

Do you want your identity to be public for this peer review? For information about this choice, including consent withdrawal, please see our Privacy Policy.

Reviewer #1: Yes: Howard Cajucom-Uy

Reviewer #2: No

---

## [Editor Report · Decision Letter 2]

5 May 2020

Corneal donation for research versus for transplantation: a-year prospective study of acceptance rates in a French University Hospital

PONE-D-19-35417R2

Dear Dr. GARCIN,

We are pleased to inform you that your manuscript has been judged scientifically suitable for publication and will be formally accepted for publication once it complies with all outstanding technical requirements.

With kind regards,

Yu-Chi Liu, M.D

Academic Editor

PLOS ONE
---

## [Editor Report · Acceptance letter]

6 May 2020

PONE-D-19-35417R2 

Corneal donation for research versus for transplantation: a-year prospective study of acceptance rates in a French University Hospital 

Dear Dr. Garcin:

I am pleased to inform you that your manuscript has been deemed suitable for publication in PLOS ONE. Congratulations! Your manuscript is now with our production department. 

With kind regards,

on behalf of

Dr. Yu-Chi Liu 

Academic Editor

PLOS ONE